# Characterization of the AcrIIC1 anti–CRISPR protein for Cas9–based genome engineering in *E. coli*

Despoina Trasanidou[1], Ana Potocnik[1,4], Patrick Barendse [1,4], Prarthana Mohanraju[1], Evgenios Bouzetos [1], Efthymios Karpouzis[1], Amber Desmet[1], Richard van Kranenburg[1,2], John van der Oost [1], Raymond H. J. Staals [1✉] & Ioannis Mougiakos [1,3✉]

Anti-CRISPR proteins (Acrs) block the activity of CRISPR-associated (Cas) proteins, either by inhibiting DNA interference or by preventing crRNA loading and complex formation. Although the main use of Acrs in genome engineering applications is to lower the cleavage activity of Cas proteins, they can also be instrumental for various other CRISPR-based applications. Here, we explore the genome editing potential of the thermoactive type II-C Cas9 variants from *Geobacillus thermodenitrificans* T12 (ThermoCas9) and *Geobacillus stearothermophilus* (GeoCas9) in *Escherichia coli*. We then demonstrate that the AcrIIC1 protein from *Neisseria meningitidis* robustly inhibits their DNA cleavage activity, but not their DNA binding capacity. Finally, we exploit these AcrIIC1:Cas9 complexes for gene silencing and base-editing, developing Acr base-editing tools. With these tools we pave the way for future engineering applications in mesophilic and thermophilic bacteria combining the activities of Acr and CRISPR-Cas proteins.

[1] Laboratory of Microbiology, Wageningen University and Research, Wageningen, The Netherlands. [2] Corbion, Gorinchem, The Netherlands. [3] SNIPR Biome, Copenhagen, Denmark. [4] These authors contributed equally: Ana Potocnik, Patrick Barendse. ✉email: raymond.staals@wur.nl; ymouyiakos@googlemail.com

Clustered Regularly Interspaced Short Palindromic Repeats (CRISPR) and CRISPR-associated (Cas) proteins are part of prokaryotic adaptive immune systems. Several Cas nucleases have been repurposed as useful tools for genome editing and transcriptional control[1,2]. CRISPR-Cas systems are divided into 2 classes, 6 types, and >30 subtypes based on their signature Cas genes. Class 2 systems have been widely used as genetic engineering tools, due to their streamlined architecture of a single effector protein (i.e. Cas9, Cas12 and Cas13), rather than a multiprotein effector complex, as is the case for Class 1 systems[3]. Despite the numerous applications of Cas12 (type V)[4] and Cas13 (type VI)[5] systems, the most extensive toolbox is based on Cas9 (type II) proteins. The type II-A Cas9 from *Streptococcus pyogenes* strain SF370 (SpyCas9)[6] is the best characterized CRISPR-Cas variant to date.

The Cas9-based genome editing technology relies on a Cas9 endonuclease and a target-specific single-guide RNA (sgRNA)[6]. The sgRNA allows the Cas9 nuclease to bind to a target DNA sequence (protospacer). After recognition of a matching target sequence, a double-stranded DNA break (DSDB) will be introduced. The protospacer is complementary to the 5'-end of the sgRNA (spacer) and flanked downstream by a short conserved motif (protospacer adjacent motif, PAM)[7,8]. Cas9-mediated DSDBs are lethal for most prokaryotes, due to the scarcity of non-templated DNA repair mechanisms, including Non-Homologous End Joining and Alternative End Joining[9]. Hence, an exogenous DNA template is often required for surviving a DSDB, using either the host's homologous recombination (HR) machinery, or heterologously-expressed (phage) recombinases (e.g. lambda-red)[10–12]. The typical low efficiency of native HR-mediated repair mechanisms from the host is the reason why Cas-mediated DSDBs are predominantly used as a counter-selection system, i.e. to kill the unedited cells after phage-recombinase-assisted HR[13,14]. A practical drawback is that this strategy often requires multiple plasmids and/or strictly controlled promoters[15–18].

To overcome this limitation, CRISPR-mediated base-editing has been developed by fusing Cas9 to a deaminase that modifies a single nucleotide base at the target site. Base-editing is used for introducing site-specific mutations, or for gene disruption by generation of premature stop codons. As base-editors generally introduce single strand nicks (not DSDBs) or no nicks at all, there is no need for efficient recombination machineries, repair DNA templates, and other exogenous factors[19]. The first developed CRISPR-based base-editing systems are "Target-AID" (activation-induced cytidine deaminase)[20] and "BE" (base editor)[21], which have also been applied in eukaryotes. They comprise a fusion between SpyCas9 (either the catalytically dead variant (dSpyCas9) or a nickase (nSpyCas9)) and a cytidine deaminase enzyme (converting a C•G into a T•A base pair). The "Target-AID" system employs the *Petromyzon marinus* cytidine deaminase (PmCDA1) or its human ortholog (human AID), while the "BE" system applies the rat APOBEC1 (rAPOBEC1). Currently reported base-editors mostly rely on the type II-A SpyCas9 variant, which has pros and cons. First, SpyCas9 enables editing at a very narrow window located at the PAM-distal end of a targeted protospacer, which is good for precise editing, but not in case of gene disruption. Second, the protospacer must be flanked immediately downstream by an 5'-NGG-3' PAM, limiting the targetable sites[20,21]. Third, SpyCas9 is only active in vivo at temperatures below 42 °C[22], restricting base-editing exclusively to mesophiles. Since these restrictions may narrow the flexibility and applicability of base-editors, systems based on alternative CRISPR-Cas variants have recently attracted special attention[23–26].

The relatively underexplored type II-C Cas9 proteins have gained substantial interest because of their small size, high fidelity, variable PAM preferences, activity even at harsh conditions (human plasma, high temperatures/salt concentrations), high flexibility with regards to interacting with different sgRNAs, and off-switch control by anti-CRISPR proteins (Acrs) with unique inhibition mechanisms[27,28]. The more compact recognition (REC) lobe of II-C Cas9 proteins is probably responsible for their weaker dsDNA unwinding activity, their reduced dsDNA binding affinity and stability, as well as their lower dsDNA cleavage activity[28]. Although these properties may decrease the on-target efficiency, these variants exhibit limited off-targeting compared to SpyCas9. This appealing feature is additionally supported by the natural ability of the II-C Cas9 proteins to recognize longer target sequences and PAMs that minimize editing at undesired sites, preventing toxic off-target effects[29,30]. Despite these attractive properties of II-C CRISPR nucleases, their use for genome editing applications has remained largely unexplored.

Acrs are small, phage-encoded proteins that evolved to inhibit CRISPR-Cas systems during the arms-race between phages and their prokaryotic hosts[31]. Acrs are being discovered and characterized at a rapidly increasing rate since their initial discovery, and a great arsenal of Acr mechanisms has already been established for various CRISPR-effectors including II-C Cas9 ortholog[32]. Nonetheless, Acrs have rarely been exploited in genome engineering applications for a function other than their ability to obstruct the DNA binding activities of CRISPR-Cas nucleases[31,33]. It was recently demonstrated that AcrIIC1 from *Neisseria meningitidis* binds the HNH domain of many II-C Cas9s, blocking target DNA cleavage (both in vitro and in vivo) while still allowing the binding to the DNA target (shown only in vitro)[34,35]. These findings provide opportunities to use this family of Acrs in combination with Cas9 variants not only for inhibition of (excessive) Cas9 cleavage activity during genome editing, as reported for most Acrs, but also as an alternative to catalytically inactive Cas9 variants in gene silencing and base-editing applications.

Here, we selected two II-C Cas9 orthologs with a wide-temperature range (20 to 70 °C) from *Geobacillus thermodenitrificans* T12 (ThermoCas9)[36] and *Geobacillus stearothermophilus* (GeoCas9)[37] to study their DNA cleaving and binding activities, as well as their potential for genome editing, transcriptional silencing and base-editing in *E. coli*. These Cas9 variants were selected for their small size and their high stability in harsh conditions, making them suitable candidates for diverse in vitro and in vivo applications. In addition, we characterized the ability of AcrIIC1 to inhibit the DNA cleavage activity of ThermoCas9 and GeoCas9 in vivo, providing an 'off-switch' for genome editing applications. Moreover, we examined the effect of AcrIIC1 on the in vivo DNA binding stability of these Cas9 nucleases, and we developed Class 2 CRISPR-Acr tools for gene silencing and base-editing. Altogether, we expand the genetic engineering toolbox.

## Results

**AcrIIC1 inhibits in vivo DNA cleavage by ThermoCas9 and GeoCas9 in *E. coli*.** Due to the thermophilic nature of the selected Cas nucleases, we first evaluated the in vivo DNA cleavage activity of ThermoCas9 and GeoCas9 in *E. coli*, at 37 °C. For this purpose, we employed an *E. coli* DH10b strain with a genome-integrated and constitutively expressed *gfp* gene (*E. coli_gfp*; Supplementary Table 1). We constructed targeting ThermoCas9 and GeoCas9 plasmids, namely pTCas9 and pGCas9 respectively, with expression of the *cas9* genes under the control of a synthetic, IPTG-inducible promoter, while opting for constitutive expression of the sgRNA module (Fig. 1a;

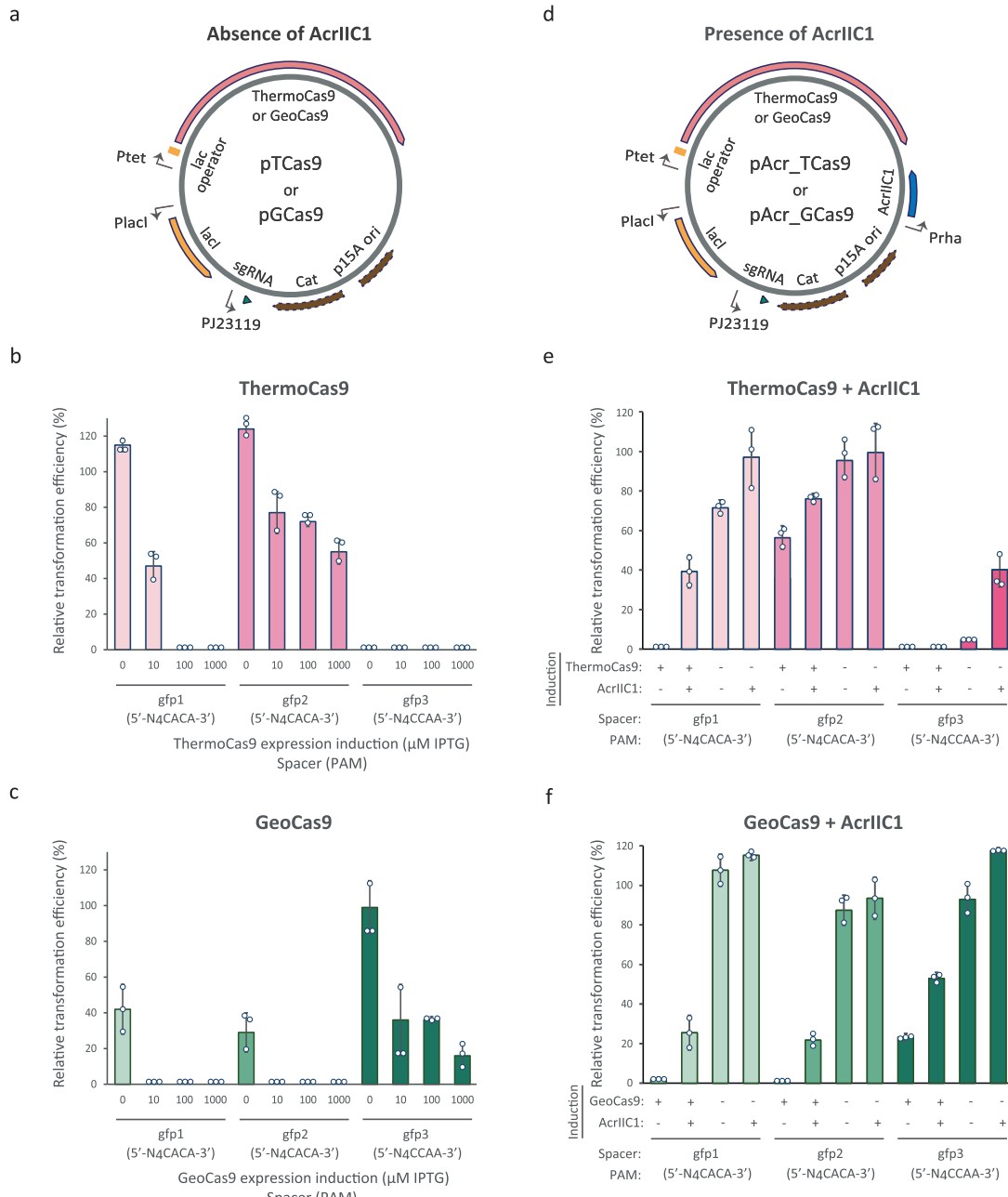

**Fig. 1 Cleavage activity of ThermoCas9 and GeoCas9 in vivo (37 °C) and inhibition by AcrIIC1. a** Schematic illustration of the construct transformed into the *E. coli_gfp* strain for killing assays. **b**, **c** Relative to the non-targeting spacer transformation efficiencies for the killing assays with ThermoCas9 and GeoCas9. The expression of the Cas9 proteins was induced using increasing IPTG concentrations (0, 10, 100, 1000 µM). **d** Schematic illustration of the construct transformed for killing-inhibition assays. **e**, **f** Relative to the non-targeting spacer transformation efficiencies for the killing-inhibition assays with ThermoCas9 and GeoCas9. The plus (+) and minus (-) symbols represent induction (1000 µM IPTG for Cas9; 0.2% w/v L-rhamnose for AcrIIC1) and absence of induction of protein expression, respectively. Bar graphs were created based on results from three independent biological replicates shown as circles. Error bars represent the standard deviation.

Supplementary Data 1). We used both nucleases to target the same three protospacers: two in the promoter region (gfp1 and gfp2) and one in the coding region of the *gfp* gene (gfp3) (Supplementary Fig. 1; Supplementary Data 2). Protospacers gfp1 and gfp2 had a 5′-N$_4$CACA-3′ PAM, while protospacer gfp3 had a 5′-N$_4$CCAA-3′ PAM. Both PAM sequences have been shown to allow ThermoCas9 cleavage in vitro[36], with varying degrees of preference at 37 °C. A previous study[37] in silico predicted and experimentally validated the 5′-N$_4$CRAA-3′ motif as the GeoCas9 PAM. Nonetheless, to gain more insight in the

PAM preference of GeoCas9, we tested the nuclease activity on protospacers gfp1, gfp2 and gfp3. We transformed all the targeting plasmids in the *E. coli_gfp* strain. In most cases, the relative transformation efficiency of the *E. coli_gfp* strain was remarkably reduced at high Cas9 induction conditions (Fig. 1b, c). ThermoCas9 could efficiently target the protospacer with the 5′-N$_4$CCAA-3′ PAM (gfp3), and to a lesser degree one of the protospacers with the 5′-N$_4$CACA-3′ PAM (gfp1) (Fig. 1b). In contrast, GeoCas9 could efficiently target both protospacers with the 5′-N$_4$CACA-3′ PAM (gfp1 and gfp2), and more weakly the

protospacer gfp3 (Fig. 1c). Overall, this strongly suggests that ThermoCas9 and GeoCas9 induce lethal DSDBs at 37 °C and they have distinct PAM- and spacer-dependent cleavage activities regardless of their high identity (87% on amino acid level).

Recent in vitro studies on the AcrIIC1 protein from *Neisseria meningitidis* reported that AcrIIC1 binds to the HNH domains of various II-C and some II-A Cas9 endonucleases, blocking their DNA cleavage activity but not affecting their DNA binding ability[35,38–40]. Prompted by the potential to use this mode of Acr inhibition for a versatile regulatory tool for CRISPR-based applications, we studied the potential of AcrIIC1 to inhibit the in vivo DNA cleavage activities of ThermoCas9 and GeoCas9. We set the plasmid-based expression of the *acriic1* gene under the control of the L-rhamnose-inducible promoter ($P_{rha}$) and transformed the resulting pAcr construct into the *E. coli_gfp* strain (Supplementary Fig. 2a; Supplementary Data 1). We subsequently transformed the resulting *E. coli_gfp*: pAcr strain with the pTCas9 and pGCas9 plasmids targeting the three *gfp* protospacers (Supplementary Fig. 2a; Supplementary Data 1, 2). In this two-plasmid approach, AcrIIC1 robustly inhibited the cleavage activity of both nucleases (Supplementary Fig. 2b, c). Even uninduced/leaky AcrIIC1 expression was enough to restore the transformation efficiencies for the most efficiently targeted protospacers (gfp1 and gfp3 for ThermoCas9, gfp1 and gfp2 for GeoCas9), due to the high copy number of pAcr and the medium copy number of pTCas9 or pGCas9. Therefore, in an alternative approach, we introduced the AcrIIC1 expressing module in the pTCas9 and pGCas9 targeting plasmids, constructing the pAcrTCas9 and pAcrGCas9 series of plasmids (Fig. 1d; Supplementary Data 1). We repeated the killing assays with the *E. coli_gfp* strain and, also with this approach, the cleavage activities of the nucleases were generally obstructed (Fig. 1e, f). The reduced transformation efficiencies for the most efficient spacers were alleviated when we induced the AcrIIC1 expression while keeping the expression of the nucleases uninduced (Fig. 1e, f). Overall, we developed AcrIIC1:Cas9 expression systems in which AcrIIC1 blocks, either constitutively or inducibly, the in vivo DNA cleavage activities of ThermoCas9 and GeoCas9 in *E. coli*.

**AcrIIC1 blocks ThermoCas9- and GeoCas9-based genome editing in *E. coli*.** After establishing that ThermoCas9 and GeoCas9 can provide strong counter-selective pressure in *E. coli* (Fig. 1b, c), we continued by evaluating the potential of these nucleases for homologous recombination (HR)-based genome editing, by combining λ-Red recombineering[41] with Cas9-mediated counter-selection. As a proof of principle, we set out to delete the *gfp* gene from the genome of *E. coli_gfp*. We selected six *gfp* protospacers per nuclease, with varying PAM preferences[36,37], and we incorporated the expression cassettes of the corresponding spacers into the pTCas9 and pGCas9 plasmids (Supplementary Data 2). In the same plasmids, we cloned the genomic regions upstream and downstream of *gfp* as the donor template for HR, resulting in the pHR_TCas9 and pHR_GCas9 series of editing plasmids (Supplementary Fig. 3; Supplementary Data 1). We transformed the *E. coli_gfp* strain with a plasmid encoding the λ-Red recombineering machinery (pKD46; Supplementary Table 1, Supplementary Data 1) and subsequently with the pHR_TCas9 or pHR_GCas9 plasmids. Finally, we screened single colonies by colony PCR and Sanger sequencing for the desired deletion. This ThermoCas9- and GeoCas9-based counter-selection approach improved the genome editing efficiency of λ-Red recombineering in *E. coli* from less than 10%[42] up to 100% (Fig. 2a, b). We obtained predominantly clean mutants (colonies with solely knock-out genotype) when we

targeted protospacers with preferred PAMs[36,37], while poor editing was observed in the case of protospacers with less preferred PAMs (Fig. 2a, b), in line with the results from the killing assays (Fig. 1b, c). Overall, the developed HR-ThermoCas9 or -GeoCas9 counter-selection systems are as efficient as other previously reported Cas9-based editing tools in *E. coli*[15,17,43–47].

We subsequently examined the effect of the AcrIIC1 expression on the HR-ThermoCas9 and GeoCas9 counter-selection editing systems in order to examine the potential of AcrIIC1 as an off-switch in genome editing applications. We introduced the AcrIIC1 expressing module into the editing constructs, generating the pHR_AcrTCas9 and pHR_AcrGCas9 constructs (Supplementary Data 1; Supplementary Fig. 4), and repeated the editing experiments. AcrIIC1 greatly blocked ThermoCas9- and GeoCas9-based counter-selection, resulting in lower editing and transformation efficiencies (Fig. 2c, d; Supplementary Fig. 5). Mixed mutants (colonies with both knock-out and wild-type phenotypes) were mainly detected for protospacers with preferred PAMs[36,37], while basal or no editing was observed in the case of protospacers with less preferred PAMs (Fig. 2c, d). Overall, we demonstrate that AcrIIC1 reduces or completely blocks editing when combining recombineering with ThermoCas9- or GeoCas9-based counter-selection.

**AcrIIC1:Cas9 complexes mediate successful transcriptional regulation in *E. coli*.** It was previously demonstrated in vitro that AcrIIC1 does not impede the binding of the II-C Cas9 nuclease from *Neisseria meningitidis* to its DNA target[35]. To investigate whether AcrIIC1-mediated inhibition of Cas9 could be harnessed for transcriptional regulation, we examined the impact of AcrIIC1:ThermoCas9 and AcrIIC1:GeoCas9 complex formation on DNA binding efficiencies in vivo and compared them to the DNA binding efficiency of the nuclease deficient ("dead") ThermoCas9 and GeoCas9 variants (dThermoCas9 and dGeoCas9, respectively). For this purpose, we introduced mutations that disrupt the active sites of the RuvC and the HNH domains (ThermoCas9$^{D8A,H582A}$; GeoCas9$^{D8A,H582A}$), generating the pdTCas9 and pdGCas9 plasmids (Supplementary Data 1). We then compared the GFP expression of the *E. coli_gfp*: pdTCas9/ pdGCas9 strains to the corresponding six *E. coli_gfp*: pAcr + pTCas9/pGCas9 strains we constructed for the killing-inhibition assays (Supplementary Fig. 2a). We found that dThermoCas9, dGeoCas9, AcrIIC1:ThermoCas9 and AcrIIC1:GeoCas9 can efficiently downregulate GFP expression (Supplementary Fig. 6). Although the silencing capabilities of the AcrIIC1:ThermoCas9 and AcrIIC1:GeoCas9 complexes were somewhat lower, these results demonstrate that AcrIIC1 can be used to uncouple the DNA binding and DNA cleaving activity of Cas9 nucleases for silencing purposes. The lower silencing capabilities of the AcrIIC1:Cas9 complexes could be attributed to the negative effect of AcrIIC1 on the expression or the stability of the nucleases it targets[39]. In addition, we cannot rule out differences between the DNA binding capacities of the active and the deactivated versions of the two nucleases (see below).

**AcrIIC1 does not hinder the in vivo DNA binding of ThermoCas9 and GeoCas9 to their targets.** Next we examined the DNA binding strength of the active nucleases (in the absence of AcrIIC1), and we compared them to their deactivated variants and AcrIIC1:Cas9 complexes. We started by studying the DNA cleavage activities of ThermoCas9 and GeoCas9 in the presence of spacer-protospacer mismatches, aiming to identify the number of mismatches that abrogates DNA cleavage. We constructed libraries of targeting plasmids with increasing numbers of 5′-end mutations for each of the aforementioned targeting spacers

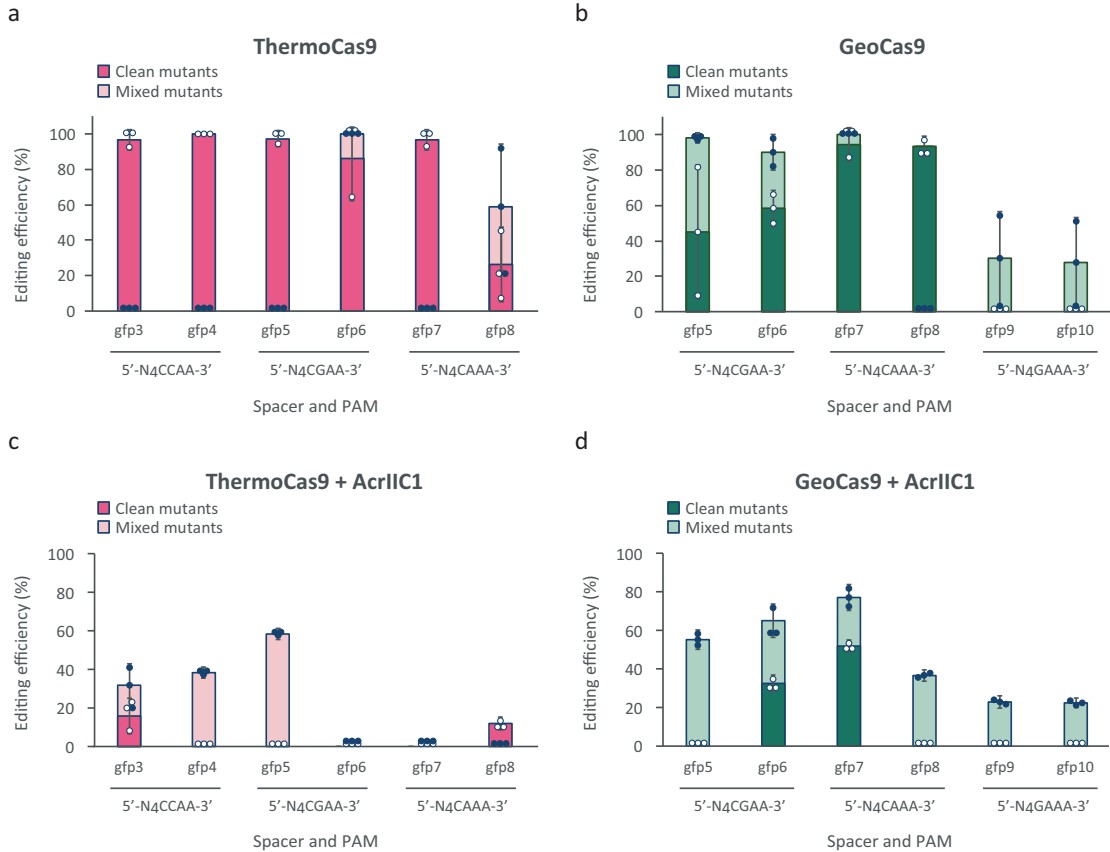

**Fig. 2 AcrIIC1-mediated inhibition of ThermoCas9- and GeoCas9-based genome editing in *E. coli*.** Editing efficiencies (%) of ThermoCas9 **a**, **c** and GeoCas9 **b**, **d** in *E. coli_gfp*, in the absence **a**, **b** or presence **c**, **d** of AcrIIC1. The editing efficiency (%) represents the % of edited versus screened single colonies, while the clean-mutant efficiency (%) represents the % of clean-mutant versus screened colonies. Dark pink and green bars indicate clean mutant editing efficiencies, while light pink and green bars represent mixed mutant editing efficiencies. Cas9 expression was fully induced in all cases. AcrIIC1 expression in **c** and **d** was fully induced. Bar graphs were created based on results from three independent biological replicates shown as white circles for clean mutants and as black circles for mixed mutants. Error bars represent the standard deviation.

(Supplementary Fig. 7; Supplementary Data 1, 2) and we repeated the transformation-based killing assays. The assays revealed that both ThermoCas9 and GeoCas9 require at least 19 bp long spacer-protospacer complementarity for measurable cleavage of some of the most efficiently targeted protospacers (gfp3 for ThermoCas9, gfp1 and gfp2 for GeoCas9) (Fig. 3a). For the less efficiently targeted protospacers (gfp2 for ThermoCas9 and gfp3 for GeoCas9) full spacer-protospacer complementarity (23 bp) was required for weak cleavage (Fig. 3a). Taken together, both ThermoCas9 and GeoCas9 require at least 19 bp spacer-protospacer complementarity to cleave DNA.

Afterwards, we compared the DNA binding capabilities of the active nucleases with spacer mismatches that prevent DNA cleavage to their deactivated variants and AcrIIC1:Cas9 complexes. We employed the *E. coli_gfp* and *E. coli_gfp*: pAcr strains respectively, and we used silencing of the expression of the *gfp* gene as a readout for DNA binding. We extended the previously constructed spacer-protospacer mismatch plasmid libraries and we kept the numbers of tested spacer-protospacer matches below 19 bp to avoid DNA cleavage when using the active nucleases (Supplementary Fig. 7; Supplementary Data 1, 2). We subsequently transformed the *E. coli* strains with the plasmid libraries and we measured the effect of each Cas9-spacer combination on the GFP expression. For all the tested strains, the reduction in GFP expression was negligible for spacer-protospacer complementarities of 8 bp or less (Fig. 3b, c). These results are in agreement with the speculated seed region for these nucleases

(PAM-proximal region of the protospacer), showing that DNA binding is strongly reduced when introducing spacer-protospacer mismatches that extend into this region[48]. Interestingly, dThermoCas9 blocked the GFP expression more efficiently than active ThermoCas9 for all tested targets and mismatch combinations (Fig. 3b). Although this observation could be attributed to differences between the expression of ThermoCas9 and dThermoCas9, Western-blot ruled out this explanation (Supplementary Fig. 8). The reverse trend, albeit far less pronounced, was observed for GeoCas9 and dGeoCas9 (Fig. 3c). These results indicate that the binding activities of ThermoCas9 and GeoCas9 cannot be linearly associated with the binding activities of their nuclease deficient counterparts. It is also noteworthy that dThermoCas9 did silence the GFP expression when guided to protospacer gfp2 (Supplementary Fig. 6a), whereas ThermoCas9 showed low DNA cleavage activity for this protospacer (Fig. 1b). Hence, the DNA cleavage activities of these nucleases do not always correlate with their binding activities. Furthermore, dThermoCas9 and dGeoCas9 showed similar binding activities for protospacer gfp2 (Supplementary Fig. 6), whereas ThermoCas9 and GeoCas9 presented different DNA cleavage activities for this protospacer (Fig. 1b, c). This suggests differences between the binding and the cleaving requirements of these highly similar Cas9 orthologs. Finally, taking into account the previously reported negative effect of AcrIIC1 on the expression or the stability of II-C Cas9 nucleases[39], we expected that the AcrIIC1:Cas9 complexes would show lower DNA binding

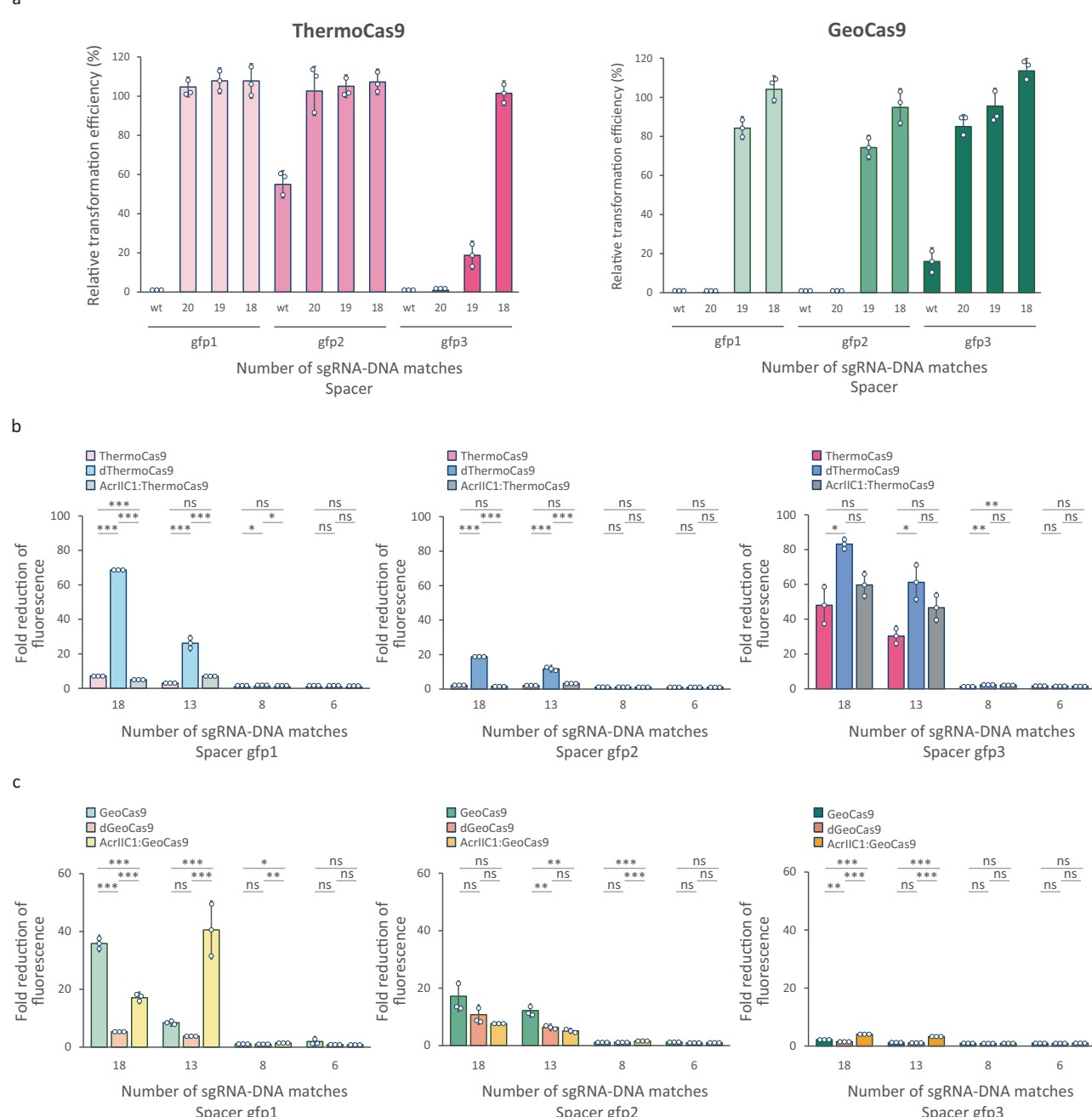

**Fig. 3 In vivo cleavage and binding specificity of ThermoCas9 and GeoCas9, and comparison to their deactivated variants and AcrIIC1:Cas9 complexes. a** Transformation efficiencies of *E. coli_gfp* cells with constructs that express ThermoCas9 (left) or GeoCas9 (right) and sgRNAs with decreasing number of matches to the targeted protospacers (wt = 23 nt). **b**, **c** Fluorescence reduction assays of *E. coli_gfp* cells that express sgRNAs with decreasing number of matches to the targeted protospacers and **b** ThermoCas9 (pink), dThermoCas9 (blue) or AcrIIC1:ThermoCas9 (grey); **c** GeoCas9 (green), dGeoCas9 (orange) or AcrIIC1:GeoCas9 (yellow). Bar graphs were created based on results from three independent biological replicates shown as circles. Error bars represent the standard deviation. Statistical significance was calculated with Pairwise Welch's t-tests and the Benjamini & Hochberg *p*-value adjustment. *p* > 0.05 is shown as ns, *p* < 0.05 is shown as *, *p* < 0.01 is shown as ** and *p* < 0.001 is shown as ***.

efficiencies compared to the sole Cas9s. Nonetheless, the DNA binding activities of the AcrIIC1:ThermoCas9 and AcrIIC1:Geo-Cas9 complexes were generally similar to the sole ThermoCas9 and GeoCas9 (Fig. 3b, c). Hence, our results demonstrate that AcrIIC1 does not hinder the in vivo binding of ThermoCas9 and GeoCas9 to their DNA targets. Moreover, differences in DNA binding abilities of the AcrIIC1:Cas9 complexes and the dCas9 proteins originate from innate differences in DNA binding capacities of the active and deactivated variants.

**AcrIIC1:Cas9 complexes as alternatives of dThermocas9 and dGeocas9 for base-editing in *E. coli*.** We continued by developing dThermoCas9- and dGeoCas9-associated base-editors, and by studying their editing outcomes. For this purpose, we fused the *Petromyzon marinus* cytidine deaminase (PmCDA1)[20] gene and the phage PBS2 uracil DNA glycosylase inhibitor (UGI)[49] gene to the C-terminus of the *dthermocas9* and *dgeocas9* genes (Supplementary Fig. 9; Supplementary Data 1). We cloned six spacers into the resulting dThermoTarget-AID and dGeoTarget-AID

plasmids, designed to target protospacers within the *gfp* gene of the *E. coli_gfp* strain (Supplementary Data 1, 2). Four of the ThermoCas9 spacers (gfp4, gfp5, gfp7, gfp8) and three of the GeoCas9 spacers (gfp5, gfp7, gfp8) were the same spacers we previously used for the corresponding HR-Cas9 counter-selection editing experiments (Fig. 2a, b; Supplementary Data 1, 2). These protospacers were selected to, collectively, contain cytidines at almost all the positions (Supplementary Data 2), allowing a fair assessment of the editing windows of the base-editors. We transformed the dThermoTarget-AID and dGeoTarget-AID plasmids in the *E. coli_gfp* strain and screened single colonies for base-editing activity by sequencing PCR amplicons spanning the targeted regions in the *gfp* gene. We detected C•G to T•A conversions at multiple positions for all the protospacers we targeted with the dThermoTarget-AID system and for four out of the six protospacers we targeted with the dGeoTarget-AID system. However, most of the edited colonies had mixed genotypes and only a few had a clean conversion at one or more positions (Fig. 4a, b; Supplementary Fig. 10). Up to two and four simultaneous clean mutations were generated by the dThermoTarget-AID and dGeoTarget-AID systems, respectively (Supplementary Fig. 10). Both base-editing systems preferentially edited cytidines at the PAM-distal end of the protospacer (Supplementary Fig. 10), similar to the commonly used SpyCas9 base-editors[50]. The observed editing windows in many occasions were extended outside of the protospacer region and were up to 28 bp for dThermoTarget-AID (from −6 to −33 positions relative to the PAM) and up to 15 bp for dGeoTarget-AID (from −10 to −24 positions relative to the PAM) (Supplementary Fig. 10). These editing windows are larger than the 5 bp (from −16 to −20 positions relative to the PAM) activity window reported for the SpyCas9 base-editors[50,51]. Interestingly, protospacer gfp8 was efficiently edited by both base-editors (Fig. 4a, b), whereas the HR-based genome editing of gfp8 by the ThermoCas9 counter-selection system was rather inefficient (Fig. 2a, b). A similar case is protospacer gfp2, for which high binding strength of dThermoCas9 and low cleavage activity of its active nuclease were observed (Fig. 1b; Supplementary Fig. 6a). In other words, a protospacer that is efficiently cleaved is not necessarily a good target for efficient base-editing. Vice versa, the targeting of the gfp5 and gfp7 protospacers resulted in low base-editing but high HR-editing outcomes (Figs. 2a, b and 4a, b). These results show that the steady binding of the base-editing complexes to their targets does not ensure high editing efficiencies, suggesting that the conversion process is governed by additional rules. Streaking and incubation of randomly selected colonies resulted in the editing of previously unedited protospacer positions, and the further extension of the editing windows (up to 30 bp for dThermoTarget-AID and 25 bp for dGeoTarget-AID), predominantly towards the PAM proximal region (Fig. 4e). Moreover, the number of simultaneous clean C•G to T•A conversions was increased to up to eight for the dThermoTarget-AID system and up to six for the dGeoTarget-AID system (Supplementary Fig. 10). Hence, the base-editing efficiency and window of activity of these dThermoCas9/dGeoCas9-associated editors can be further increased by prolonging the editing conditions. The dThermoTarget-AID and dGeoTarget-AID systems were additionally applied for base-editing at three different endogenous sites (*pyrE*, *xylB* and *adhE* genes), presenting similar characteristics to those observed for the *gfp* targets (Supplementary Data 1, 2; Supplementary Figs 11a, b, 12).

Furthermore, we reasoned to combine the DNA binding activities of the AcrIIC1:Cas9 complexes with the base-editing activity of the PmCDA1 enzyme. We expected that the resulting systems can not only be an alternative to the "dThermoTarget-AID" and "dGeoTarget-AID" systems, but also have the additional benefit of being able to select against unedited loci by relieving the Acr-mediated inhibition of DNA cleavage activity of Cas9. Specifically, these Acr base-editors would first induce base-editing of the target site by expressing AcrIIC1 that blocks the Cas9 nuclease activity of the Cas9-PmCDA1 fusion protein and allows for PmCDA1-mediated deamination of the target base(s). Subsequent interruption of the AcrIIC1 expression would result in Cas9-mediated counter-selection of the wild-type cells. In addition, this could enrich colonies with clean mutations. We introduced the AcrIIC1 expressing module in the Target-AID plasmids and we substituted the *dcas9* genes with their catalytically active counterparts, developing the "AcrThermoTarget-AID" and "AcrGeoTarget-AID" base-editing systems (Supplementary Fig. 9; Supplementary Data 1). We transformed the constructed plasmids in the *E. coli_gfp* strain, performed *gfp*-specific PCRs on single colonies, and sequenced the products. As expected, the percentage of clean point mutations in protospacers that were moderately or highly edited was substantially higher compared to the corresponding number from the "dThermoTarget-AID" and "dGeoTarget-AID" systems (Fig. 4a–d; Supplementary Fig. 10). Moreover, we detected an increased preference for editing at the PAM-proximal ends when compared to the dCas9-based editors (Supplementary Fig. 10). This could be attributed to the tolerance of these active Cas9s in mismatches at the PAM-distal end, which results in counter-selection of cells with this mutant genotype (Fig. 3a). In contrast, mismatches at the PAM-proximal end are not tolerated, blocking counter-selection and allowing for cell survival (Fig. 3a). The observed editing windows were up to 15 bp for AcrThermoTarget-AID (from −9 to −23 positions relative to the PAM) and up to 22 bp for AcrGeoTarget-AID (from −3 to −24 positions relative to the PAM), while the number of simultaneous clean mutations was up to four and three, respectively (Supplementary Fig. 10). Streaking and incubation of randomly selected colonies resulted in the editing of previously unedited protospacer positions, the extension of the activity window (in the case of AcrThermoTarget-AID) and in colonies with predominantly clean mutations at all the edited positions (Fig. 4e; Supplementary Fig. 10). Last, the AcrThermoTarget-AID and AcrGeoTarget-AID systems were additionally applied for base-editing at the endogenous *pyrE*, *xylB* and *adhE* sites, exhibiting similar traits to those observed for the *gfp* sites (Supplementary Data 1, 2; Supplementary Figs. 11c, d, 12).

Overall, these type II-C editors mediate efficient base-editing in wide activity windows and multiple positions simultaneously, facilitating gene inactivation in bacteria in the absence of donor DNA. The Acr-mediated control over Cas9 cleavage activity additionally enables counter-selection of non-edited loci, further increasing the number of clean mutants and the preference for editing at the PAM-proximal protospacer region.

## Discussion

In this study, we initially demonstrate that the thermostable type II-C ThermoCas9 and GeoCas9 endonucleases are highly active in vivo at 37 °C and introduce lethal DSDBs in the *E. coli* genome. Moreover, we reveal additional PAM recognition sequences of GeoCas9, for which very few PAMs have been reported earlier[37]. In agreement with our previous in vitro study[36], we show that ThermoCas9 and GeoCas9 (under these conditions) exhibit lower in vivo mismatch tolerance compared to previous data for SpyCas9[52]. The enhanced fidelity of these II-C Cas9 variants can be attributed not only to their longer PAM (4 bp) and spacer (23 bp), but also to their slower dsDNA cleavage rates (at 37 °C, lower $k_{cat}$)[28,36,53]. These characteristics are similar to the II-C Cas9 variants from *N. meningitidis* (NmeCas9), and

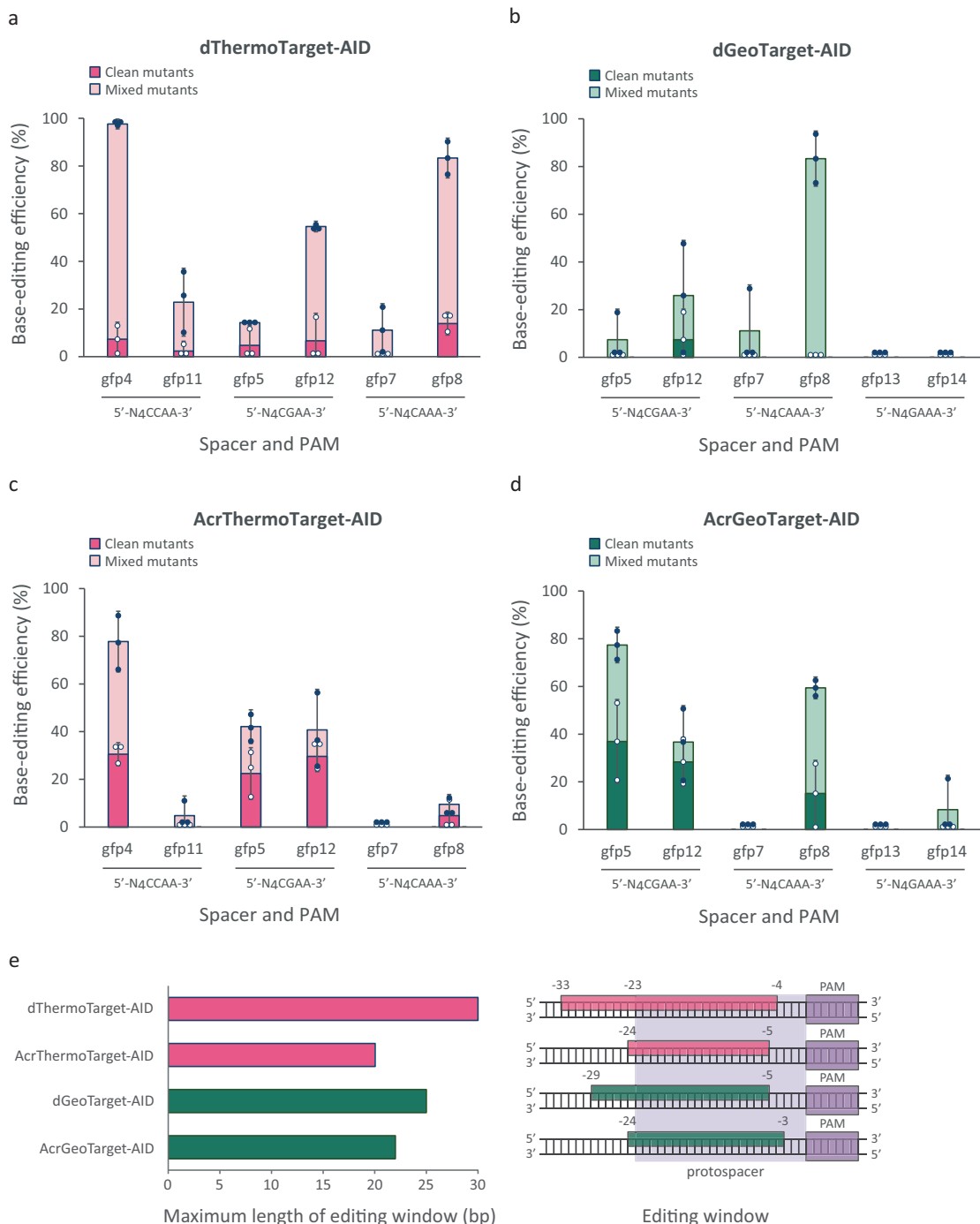

**Fig. 4 Type II-C cytidine base-editors and their AcrIIC1:Cas9 complexes that enable base-editing coupled to counter-selection in *E. coli*.** Base-editing efficiency (%) of the dThermoCas9 **a**, dGeoCas9 **b**, AcrIIC1:ThermoCas9 **c**, and AcrIIC1:GeoCas9 **d** base-editors in *E. coli_gfp*. The base-editing efficiency (%) represents the % of the number of edited versus screened single colonies, while the clean-mutant efficiency (%) represents the % of clean-mutant versus screened single colonies. Dark pink and green bars indicate clean mutant editing efficiencies, while light pink and green bars represent mixed mutant editing efficiencies. Bar graphs were created based on results from three independent biological replicates shown as white circles for clean mutants and as black circles for mixed mutants. Error bars represent the standard deviation. **e** The maximum length (left) and spectrum (right) of the editing window of each base-editing system. The dark purple boxes indicate the PAM region, while the light purple area depicts the protospacer region. Pink and green boxes represent the overall activity window of the base-editing systems, after analysis of all screened colonies either before or after streaking.

*Campylobacter jejuni* (CjeCas9), which are also more specific than SpyCas9[29,30]. Leveraging their functionality at 37 °C, we apply ThermoCas9 and GeoCas9 as counter-selection tools in combination with λ-Red recombineering in *E. coli*, leading to high editing efficiencies and predominantly clean mutants. In addition, we show that the previously in vitro characterized Acr

protein from *N. meningitidis* (AcrIIC1)[34,35] robustly inhibits the DNA cleavage activity of both ThermoCas9 and GeoCas9 in vivo. In line with its reported cleavage inhibition ability, AcrIIC1 successfully blocks the Cas9-mediated counter-selection, severely dropping the editing efficiencies and eliminating the presence of clean mutants. Overall, ThermoCas9 and GeoCas9

can be used as efficient genome targeting or editing tools for in vivo applications at 37 °C with the ability of 'off-switch' control by AcrIIC1.

Next, we uncovered the ability of AcrIIC1 to trap these Cas9 nucleases in a DNA-bound and cleavage-inactive state in vivo, resulting in the silencing of the targeted *gfp* gene in the genome of *E. coli*. The silencing efficiency of the AcrIIC1:Cas9 complexes was lower than that of their dCas9 variants, probably because of inherent differences in the DNA binding ability between the active and inactive form of each nuclease that we reveal in our study. We also show that AcrIIC1 does not destabilize the in vivo DNA binding strength of ThermoCas9 and GeoCas9, and we exploit this unique feature to couple base-editing applications (by allowing AcrIIC1 expression) to subsequent counter-selection (by interrupting AcrIIC1 expression). In this context, we develop AcrIIC1:Cas9 and dCas9 base-editors of these II-C variants and we observe up to six times larger editing windows than the previously reported dSpyCas9 base-editor[51], probably due to differences in the size and the structure of the II-C and II-A variants. The observed editing windows in many occasions were expanded upstream of the protospacer region. The relatively large editing window is explained partly by the fact that these II-C orthologs recognize longer sequences (23 nt) compared to the II-A variants (20 nt), forming a slightly larger R-loop with extended single-stranded DNA region available for deamination[36,37,51]. Similar to the dSpyCas9 editors[50], the dThermoCas9 and dGeoCas9 editors mainly edit the PAM-distal end of the protospacer, which makes sense as the presence of mutations within the seed region hampers the DNA binding ability of all these Cas9s. In contrast, the Acr-editors associated with the active Cas9 variants show preference for editing at the PAM-proximal end of the protospacer, because mutations at the PAM-distal end are tolerated by ThermoCas9 and GeoCas9, resulting in cleavage leading to cell death (counter-selection); hence this explains the enrichment of the PAM-proximal edits. The extended base-editing windows of all II-C editor designs described here may increase the freedom for gene inactivation (generation of premature stop codons, mutation of start codon), for gene mutagenesis (introduction of multiple nucleotide substitutions), and, theoretically, for abrogation of regulatory sequences (regions coding for small RNAs or promoter regions) and splice sites. In addition, the applicability of our active Cas9:Acr systems for gene targeting inhibition, gene silencing and base-editing could be further expanded for the study of microbial communities, which are typically composed of non-model organisms, providing an easy and flexible way to control population dynamics. Transient expression of AcrIIC1 could be used to control the growth rate and/or productivity of certain community members, without the requirement for use of a tight expression system. Controlled reduction in the AcrIIC1 expression would eliminate these members from the community, allowing to study the effect of the population changes.

## Methods

**Bacterial strains and growth conditions**. Bacterial strains used in this study are listed in Supplementary Table 1. All *E. coli* strains were cultured in Luria-Bertani broth (LB) or on LB agar plates, supplemented when necessary with chloramphenicol (15 µg ml⁻¹), and/or ampicillin (100 µg ml⁻¹). L-rhamnose (0.2% w/v), and/or IPTG (0−1000 µM) were additionally used for inducing the expression of AcrIIC1 and (d)ThermoCas9/ (d) GeoCas9/ (d)ThermoCas9-PmCDA1-UGI-LVA/ (d)GeoCas9-PmCDA1-UGI-LVA, respectively. L-arabinose (0.15% w/v) was also applied in the genome editing assays to trigger the expression of the λ-Red recombineering proteins (Exo, Beta, Gamma). Moreover, M9TG (11.28 g 1X M9 salts, 10 g tryptone, 5 g

glycerol) medium was used instead of LB in the fluorescence loss assays. All strains were grown at 37 °C (220 rpm when liquid culture), except for *E. coli_gfp*: pKD46 (Supplementary Table 1) which was grown at 30 °C.

**Construction of plasmids**. Plasmids, primers or oligonucleotides, *cas9* or *acriic1* gene and protein sequences used in this study are presented in Supplementary Data 1, 3, 4, and 5, respectively. Thanks to the high (94%) nucleotide identity between the ThermoCas9 and GeoCas9 sgRNA modules (with the few differences predicted to be part of the sgRNA loops), we used the ThermoCas9 sgRNA module for all created constructs. The bacterial plasmids were constructed using the NEBuilder HiFi DNA Assembly Cloning Kit (NEB). The fragments for assembling the plasmids were obtained through PCR with Q5® High-Fidelity 2X Master Mix (NEB). The PCR products were run on a 0.8% agarose gel and were subsequently purified using Zymogen gel DNA recovery kit (Zymo Research). The assembled plasmids were transformed to chemically competent *E. coli* DH5α cells[54] (Supplementary Table 1) and plated on LB agar containing chloramphenicol (15 µg ml⁻¹) or ampicillin (100 µg ml⁻¹) and incubated overnight at 37 °C. The next day, single colonies were inoculated in LB medium, grown overnight at 37 °C (220 rpm) and the plasmids were isolated using the GeneJet plasmid Mini-prep kit (ThermoFisher Scientific). All the constructs were verified using Sanger sequencing (Macrogen). The description of the assembled fragments used for the construction of each plasmid is detailed in Supplementary Data 1. For annealing of oligos to create dsDNA used in the plasmid assembly, 4 µl oligonucleotide pairs (100 µM each) were mixed in Milli-Q water to a final volume of 100 µl, heated at 95 °C for 5 min, and slowly cooled to room temperature.

**Killing assays**. To target bacterial DNA, chemically competent *E. coli_gfp* cells were transformed[54] with equal amounts (3 ng) of *gfp*-targeting plasmid (pTCas9_gfp1-gfp3; pGCas9_gfp1-gfp3) (Fig. 1a; Supplementary Table 1; Supplementary Data 1, 2). We also constructed libraries of mismatched targeting plasmids, namely pTCas9_x.y and pGCas9_x.y (where x = the employed spacer gfp1, gfp2, or gfp3; y = the number of consecutive spacer-protospacer matches counting from the PAM-proximal end of the protospacer) (Fig. 1a; Supplementary Fig. 7; Supplementary Data 1, 2). As control, 3 ng of a non-targeting plasmid (pTCas9_scr; pGCas9_scr) were used (Fig. 1a; Supplementary Fig. 7; Supplementary Data 1, 2). Transformed cells were cultured on LB agar supplemented with chloramphenicol (15 µg ml⁻¹) and different IPTG concentrations (0 µM, 10 µM, 100 µM, 1000 µM IPTG) for 17 h at 37 °C. Colony forming units (CFUs) were counted after plating 100 µl of undiluted biological triplicates (from 500 µl recovery) and used for calculating the relative transformation efficiencies.

**Killing-inhibition assays**. To inhibit targeting of the bacterial genomic DNA using a two-plasmid approach, *E. coli_gfp* cells harboring an AcrIIC1-expressing plasmid (pAcr) were transformed[54] with equal amounts (5 ng) of *gfp*-targeting plasmid (pTCas9_gfp1-gfp3; pGCas9_gfp1-gfp3) (Supplementary Fig. 2a; Supplementary Table 1; Supplementary Data 1, 2). As control, 5 ng of a non-targeting plasmid (pTCas9_scr; pGCas9_scr) were used (Supplementary Fig. 2a; Supplementary Data 1, 2). Transformed cells were grown on LB agar supplemented with chloramphenicol (15 µg ml⁻¹), ampicillin (100 µg ml⁻¹), IPTG (0 µM or 1000 µM), and L-rhamnose (0% or 0.2% w/v) for 17 h at 37 °C. Colony forming units (CFUs) were counted after plating 100 µl

undiluted biological triplicates and used for calculating the relative transformation efficiencies.

Alternatively following a single-plasmid approach, *E. coli_gfp* cells were transformed with equal amounts (5 ng) of targeting inhibition plasmid additionally carrying the *acriic1* gene (pAcr_TCas9_gfp1-gfp3; pAcr_GCas9_gfp1-gfp3) (Fig. 1d; Supplementary Data 1, 2). As control, 5 ng of a non-targeting plasmid (pAcr_TCas9_scr; pAcr_GCas9_scr) were used (Fig. 1d; Supplementary Data 1, 2). Transformed cells were grown on LB agar supplemented with chloramphenicol (15 μg ml$^{-1}$), IPTG (0 μM or 1000 μM), and L-rhamnose (0% or 0.2% w/v) for 17 h at 37 °C. CFUs were counted after plating 100 μl undiluted biological triplicates and used for calculating the relative transformation efficiencies.

**Genome editing assays.** For the deletion of the genomic *gfp* gene and its promoter (P$_{lacUV5}$), chemically competent *E. coli_gfp* cells harboring the λ-Red operon (*exo*, *beta*, and *gam* genes)[41] were transformed[54] with 5 ng of genome editing plasmid (pHR_TCas9_gfp3-gfp8; pHR_GCas9_gfp5-gfp10) (Supplementary Fig. 3a; Supplementary Table 1; Supplementary Data 1, 2). The λ-Red operon was transcribed from the arabinose-inducible promoter (P$_{araB}$)[55] on a thermo-sensitive, low copy number plasmid (pKD46)[56]. During recovery, cells grew for 2 and half hours at 30 °C (220 rpm) and the expression of both the λ-Red recombineering system and the Cas9 nuclease was induced (0.15% w/v L-arabinose and 1 mM IPTG, respectively) for maximum HR efficiency. The induction of the Cas9 expression was prolonged for counter-selection of the non-edited cells by plating 100 μl of undiluted biological triplicates on LB agar supplemented with chloramphenicol (15 μg ml$^{-1}$) and IPTG (1000 μM). After 17 h of incubation at 37 °C, several colonies were screened for genome editing through OneTaq® 2X Master Mix with Standard Buffer (NEB) PCR amplification of the targeted region with genome specific primers (Supplementary Data 3), and the size of the PCR products was verified by 1.2% agarose gel electrophoresis. The results were analyzed with GelAnalyzer 19.1 and colonies were divided into 3 distinct categories: (a) wild-type, (b) mixed wild-type and knockout, and (c) clean knock-out. Sanger sequencing was indicatively performed to verify successful gene deletion in the resulting mutants. Bacterial cells either lacking (*E. coli* DH10b cells) or containing the targeted fragment (*E. coli_gfp*) (Supplementary Table 1) were used as positive and negative genome editing control, respectively.

**Counter-selection inhibition assays.** To inhibit the Cas9-mediated counter-selection of unedited cells after HR, *E. coli_gfp*: pKD46 cells were transformed with equal amounts (5 ng) of counter-selection inhibition plasmid additionally carrying the *acriic1* gene (pHR_AcrTCas9_gfp3-gfp8; pHR_AcrGCas9_gfp5-gfp10) (Supplementary Fig. 4; Supplementary Table 1; Supplementary Data 1, 2). Maintaining maximum expression of the λ-Red recombineering system (0.15% w/v L-arabinose) during recovery, we fully expressed AcrIIC1 (0.2% w/v L-rhamnose) and Cas9 (1000 μM IPTG) and plated 100 μl of undiluted biological triplicates on LB agar supplemented with chloramphenicol (15 μg ml$^{-1}$) and L-rhamnose (0.2% w/v). Screening and analysis of the editing events were performed, as described above.

**Binding assays.** To quantify the fluorescence loss, *E. coli* DH10B_gfp cells were transformed[54] with equal amounts (3 ng) of *gfp*-silencing plasmid (pdTCas9_gfp1-gfp3; pdGCas9_gfp1-gfp3) and their derivatives with partial spacer-protospacer complementarity) (Supplementary Fig. 7; Supplementary Table 1; Supplementary Data 1, 2). As positive fluorescence control, 3 ng

of empty vector (pACYC184) or non-targeting plasmid (pdTCas9_scr; pdGCas9_scr) (Supplementary Data 1, 2) were used, while as a negative fluorescence control *E. coli* DH10b cells transformed with 3 ng of the empty vector were employed (Supplementary Table 1). Post-transformation, 2 μl of the recovered cells were cultured in 198 μl M9TG containing chloramphenicol (15 μg ml$^{-1}$) in a Masterblock® 96-well deep microplate (Greiner Bio-One) for 22 h at 37 °C with vigorous shaking (900 rpm). The second day, 2 μl of overnight cultures were diluted in 198 μl M9TG containing the same antibiotic, and 2 μl of these were re-diluted in 198 μl M9TG with antibiotic and IPTG (0 μM, 10 μM, 50 μM) in a Masterblock® 96-well deep micro-plate (Greiner Bio-One), and incubated for 22 h at 37 °C with vigorous shaking (900 rpm). The third day, 2 μl of overnight cultures were diluted in 998 μl 1× PBS (8 g NaCl, 0.2 g KCl, 1.44 g Na2HPO4 2H2O, 0.24 g KH2PO4; pH=6.8) in a Masterblock® 96-well deep microplate (Greiner Bio-One). The fluorescence signal and the presence of subpopulations were examined using the Attune NxT Flow Cytometer device (Thermo Fisher Scientific) (GFP intensity 405-512/25 of at least 30,000 single cells per sample). All assays were performed in three biological replicates.

Similarly, for the AcrIIC1:Cas9 complexes, *E. coli_gfp* cells harboring the AcrIIC1-expressing plasmid (pAcr) were transformed[54] with equal amounts (5 ng) of *gfp*-targeting plasmid (pTCas9_gfp1-gfp3; pGCas9_gfp1-gfp3 and their derivatives with partial spacer-protospacer complementarity) (Supplementary Fig. 2a, 7; Supplementary Table 1; Supplementary Data 1, 2). As positive fluorescence controls, 5 ng of the empty vectors (pACYC184; pUC19) or the non-targeting plasmids (pTCas9_scr; pGCas9_scr) were used (Supplementary Data 1, 2), while as negative fluorescence control 5 ng of *E. coli* DH10B cells electroporated with 5 ng of each empty vector (pACYC184; pUC19) were applied (Supplementary Table 1). The same fluorescence loss assay protocol as above was followed, with the only difference being that ampicillin (100 μg ml$^{-1}$) and L-rhamnose (0.2% w/v) were also applied.

**Western blot assays.** To compare the expression of ThermoCas9 and dThermoCas9, *E. coli_gfp* cells were transformed with equal amounts (3 ng) of a His6-(d)ThermoCas9-encoding, non-targeting plasmid (pHis-TCas9_scr; pHis-dTCas9_scr) (Supplementary Data 1, 2). As a negative control, 3 ng of a non-targeting plasmid lacking a *his6-(d)thermocas9* gene (pscr) was used (Supplementary Data 1, 2). Transformed cells were grown overnight at 37 °C on LB agar supplemented with chloramphenicol (15 μg ml$^{-1}$). Single colonies were inoculated in 5 ml LB containing chloramphenicol (15 μg ml$^{-1}$), and cell cultures were incubated at 37 °C under shaking (220 rpm) until reaching OD600 = 0.4–0.6. After 1 h incubation on ice, the His6-(d) ThermoCas9 expression was induced with 1 mM IPTG, and the cultures were incubated overnight at 20 °C for maximum protein yields. The equivalents of 1 mL of cells at OD600 = 0.8 were centrifuged at maximum speed for 1 min. The cell pellets were resuspended in 100 μl of 1× SDS-PAGE buffer, heated at 98 °C for 10 min, centrifuged at maximum speed for 5 min, and loaded in a 10% Mini-PROTEAN® TGX™ Precast Gel (Bio-Rad). 10 μl of PageRuler™ Prestained Protein Ladder and 10 μl of blot positive control marker were loaded in the gel, which was run in a Mini−PROTEAN® Tetra Vertical Electrophoresis Cell (Bio-Rad) containing 800 ml 1× SDS running buffer at constant 20 mA for 1 h (PowerPac™ Basic Power Supply; Bio-Rad). After 4 h staining in PageBlue™ Protein Staining Solution (Thermo Scientific) under gentle shaking, washing and visualization with UV light (G:BOX Chemi XX6), the gel was transferred to the iBlot2 dry blotting machine (Invitrogen) and then to the Transfer buffer (3.03 g

Trizma-base, 14.4 g Glycine, 200 ml methanol, dH2O up to 1 l) in a big agar plate. After blocking in PBST (137 mM NaCl, 2.7 mM KCl, 10.1 mM $Na_2HPO_4$, 1.8 mM $KH_2PO_4$, 0.1% Tween-20; pH 7.4) + 3% BSA for 1 h at room temperature, the membrane was incubated overnight at 4 °C under slow rotation with 6x-His Tag Monoclonal Antibody (HIS.H8; Invitrogen) diluted 1:5,000 in the same solution. The membrane was washed with PBST and incubated for 1 h at room temperature with Goat anti-Mouse IgG (H + L) Highly Cross-Adsorbed Secondary Antibody Alexa Fluor Plus 488 (Invitrogen) diluted 1:20,000 in the same solution. After the final washing of the membrane with PBST, the blot was developed using SuperSignal™ West Pico PLUS Chemiluminescent Substrate kit (Thermo Scientific) and visualized according to the instructions.

**Base-editing assays.** To introduce single nucleotide substitution(s) in the genomic *gfp, pyrE, xylB,* and *adhE* genes, *E. coli_gfp* cells were transformed[54] with 3 ng of base-editing plasmid (pdThermoTarget-AID_gfp4/gfp5/gfp7/gfp8/gfp11/gfp12/pyrE1/pyrE2/xylB1/xylB2/adhE1/adhE2; pAcrThermoTarget-AID_gfp4/gfp5/gfp7/gfp8/gfp11/gfp12/pyrE1/pyrE2/xylB1/xylB2/adhE1/adhE2; pdGeoTarget-AID_gfp5/gfp7/gfp8/gfp12/gfp13/gfp14/pyrE3/pyrE4/xylB3/xylB4/adhE3/adhE4; pAcrGeoTarget-ID_gfp5/gfp7/gfp8/gfp12/gfp13/gfp14/pyrE3/pyrE4/xylB3/xylB4/adhE3/adhE4) (Supplementary Fig. 9; Supplementary Table 1; Supplementary Data 1, 2). The expression of the dCas9-editors was induced with the addition of 50 μM IPTG during both recovery and plating, while in the case of the Acr-editors base-editing (50 μM IPTG and 0.2% w/v L-rhamnose) and counter-selection (1 mM IPTG) conditions were separately applied during recovery and plating, respectively. After 17 h of incubation at 37 °C in the presence of chloramphenicol (15 μg ml⁻¹), several colonies were streaked on "master" selection plates with no inducers and were simultaneously screened for base-editing through PCR amplification (Q5® High-Fidelity 2× Master Mix; NEB) of the targeted region with genome-specific primers. The amplified fragments were purified (DNA clean and concentrator kit; Zymo Research) and sequenced (Sanger), followed by high-throughput in silico analysis of the results employing a variation of the on-line tool "EditR"[57], previously developed in our lab. Each base-editing experiment was performed in three biological replicates. In addition, the streaks on the "master" plates from single colonies were streaked on plates supplemented with 1 mM IPTG to screen for enhanced base-editing efficiency and/or purity, as described above.

**Statistics and reproducibility.** In this study, all experiments were conducted using three independent biological replicates. The average value and the standard deviation of the individual data points were calculated and visualized using Microsoft Excel. When applicable, statistical significance was calculated with Pairwise Welch's t-tests and the Benjamini & Hochberg p-value adjustment.

**Reporting summary.** Further information on research design is available in the Nature Portfolio Reporting Summary linked to this article.

## Data availability
All data supporting the findings of this study are available within the paper and its Supplementary Information/Data. Raw data of the figures can be found in Supplementary Data 6. Plasmid maps have been deposited in Addgene and are publicly available using the following Addgene names and IDs: pTCas9_scr (#207566), pGCas9_scr (#207567), pdTCas9_scr (#207568), pdGCas9_scr (#207569), pAcr (#207570), pAcrTCas9_scr (#207571), pAcrGCas9_scr (#207572), pdThermoTarget-AID (#207573), pdGeoTarget-AID (#207574), pAcrThermoTarget-AID (#207575) and pAcrGeoTarget-AID (#207576). Any additional data are available from the corresponding author upon reasonable request.

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

## Acknowledgements

We would like to thank Dr. Thijs Nieuwkoop for his technical assistance with the flow cytometer, Dr. Sjoerd Creutzburg for offering the *E. coli*_gfp strain, and Dr. Asimenia Gavriilidou for assisting with the statistical analysis. D.T. is financially supported by the Alexander S. Onassis Foundation grant (F ZM 083-2/2018-2019). J.v.d.O. thanks the Dutch Research Council (NWO Spinoza grant SPI 93-537, and NWO Gravitation grant 024.003.019), and the European Research Council (ERC-AdG-834279) for financial support. R.H.J.S. is supported by a VIDI grant (VI.Vidi.203.074) from NWO.

## Author contributions

D.T., R.H.J.S., and I.M. conceived this study and design of experiments. D.T., A.P., P.B., E.B., E.K., and A.D. conducted the experiments. P.M., R.v.K., J.v.d.O., R.H.J.S., and I.M. supervised this project. D.T., J.v.d.O., R.H.J.S., and I.M. wrote the manuscript.

## Competing interests

The authors declare the following competing interests: J.v.d.O. is cofounder of NTrans Technologies, and a member of the Scientific Advisory Board of NTrans Technologies and Hudson River Biotechnology. J.v.d.O. and R.H.J.S. are members of the Scientific Advisory Board of Scope Biosciences. R.v.K. and I.M. are employed by the commercial companies Corbion (Gorinchem, The Netherlands) and SNIPR Biome (Copenhagen, Denmark), respectively. D.T., J.v.d.O., R.H.J.S., and I.M. are inventors of CRISPR-Cas-related patents/patent applications. All other authors declare no competing interests.

## Additional information

**Peer review information** This manuscript has been previously reviewed in another Nature Portfolio journal. The manuscript was considered suitable for publication without further review at *Communications Biology*.

