## [Peer Review File · Communications Biology]

Reviewer #1 (Remarks to the Author):

In their manuscript, "*In vivo* characterization of the AcrIIC1 anti-CRISPR for Cas9-based genome engineering," Trasanidou and co-authors used AcrIIC1 to selectively inhibit DNA cleavage by thermostable type II-C Cas9 variants (ThermoCas9 and GeoCas9) in *E. coli* cells and exploit this property for CRISPRi as well as simultaneous base editing and clone selection.

Throughout this study, the authors employed *E. coli* strains with a genomically integrated GFP to characterize ThermoCas9 and GeoCas9. They found that both Cas9 orthologues were able to induce targeted double-strand breaks, as evidenced by cell/colony depletion. Depletion efficiency was thereby dependent on the protospacer/PAM sequence used. Next, the authors used AcrIIC1 to selectively inhibit Cas9 nuclease activity with the goal of effectively converting Cas9 into a deadCas9. Importantly, however, the efficacy of CRISPRi, as measured by inhibition of GFP expression upon CRISPR effector targeting, was rather low in samples co-expressing Cas9 and AcrIIC1 compared to conventional dCas9-mediated CRISPR. Subsequently, the authors created dCas9-AID fusions for Geo- and ThermoCas9 and used them for C:G to T:A conversion in various sequences targeted in the GFP gene. Interestingly, the authors found that the base editing window is particularly long for both dCas9-AID variants, spanning up to 22 base pairs. Finally, the authors constructed a second set of Cas9-AID fusions, this time based on nuclease-competent Cas9s. They then co-expressed AcrIIC1 to greatly reduce, though probably not completely prevent, Cas9 nuclease activity. This resulted in improved "purity" of base editing outcomes, i.e. "clean" point mutations in protospacers amplified from single colonies, and also narrowed the base editing window to PAM-proximal sites.

Overall, the manuscript is well-written and the data are convincing and, for the most part, support the claims made. However, I feel that this paper lacks sufficient novelty to warrant publication in Nature Communications. I acknowledge that there are several interesting data sets in this MS, such as the report of wide base editing windows with the GeoCas9- and ThermoCas9-AIDs. On the other hand, the central claim of the MS is the characterization of AcrIIC1 and its utility for genome engineering.

We would like to thank Reviewer #1 for their useful comments and suggestions. In our study, we demonstrate that the thermostable type II-C ThermoCas9 and GeoCas9 can be used to broaden the targeting scope of genome editing, gene silencing, and base-editing in mesophiles by recognizing alternative PAMs and creating up to six times larger base-editing windows compared to previously reported SpyCas9 base-editors. Complementing this type II-C CRISPR-Cas toolbox, we show that AcrIIC1 can be used not only as an 'off-switch' for genome editing applications (like most anti-CRISPR proteins) but also as a mediator of gene silencing and base-editing in *E. coli*. To the best of our knowledge, AcrIIC1 has never been characterized *in vivo* for gene silencing and base-editing purposes. In addition, this is the first report to date on anti-CRISPR-mediated inhibition of the ThermoCas9 activity. Hence, the novelty of our work derives from the exploitation of both the AcrIIC1 and the ThermoCas9/GeoCas9 proteins.

Regarding the first aspect, the mechanism by which AcrIIC1 inhibits type II-C Cas9s, i.e. targeting the HNH domain to selectively prevent Cas9 nuclease function but not DNA binding, has been known for many years (Pawluk et al., Cell 167, 2016). It is not surprising, in my opinion, that this mechanism holds true when the protein is expressed in *E. coli*.

Pawluk et al. (Cell 167, 2016) and more recent studies (Harrington et al., Cell 170, 2017; Song et al., Cell reports 29, 2019; Garcia et al., Cell reports, 2019; Mathony et al., Nature Chemical Biology 16, 2020) showed that AcrIIC1 blocks *in vitro* the DNA cleavage but not the DNA binding ability of several

II-C and some II-A Cas9 endonucleases. However, *in vitro* functionality of a protein does not guarantee its efficient *in vivo* activity. In this study, we demonstrated that AcrIIIC1 efficiently inhibits the cleavage activity of ThermoCas9 and GeoCas9 in *E. coli* and we developed a versatile regulatory tool for *in vivo* CRISPR-based applications (genome editing, gene silencing and base-editing). Notably, AcrIIIC1 has never been used for gene silencing and base-editing purposes.

Furthermore, the authors claim to report "the first Acr tools for base-editing applications [...]". This is not true, as, for instance, AcrIIA5 has already been used to prevent off-target editing in the context of base editing in mammalian cells (Liang et al., *Cells* 9, 2020; doi:10.3390/cells9081786).

Liang et al. (*Cell* 9, 2020) applied AcrIIA5 for inhibition of base-editing at undesired loci of the mammalian genome. In contrast, we describe the exploitation of AcrIIIC1 for the enhancement of base-editing outcomes in bacteria, for example by providing higher number of clean point mutations. For clarity, we rephrased 'the first Acr tools for base-editing applications [...]' into 'the first Acr tools for the facilitation of base-editing applications [...]'.

Moreover, AcrIIIC1 seems to be of little use in context of CRISPRi in my opinion. The data in the MS shows that dCas9 is superior for CRISPRi compared to AcrIIIC1-Cas9 co-expression, the latter of which is also more complicated from an application perspective because it involves an additional component (AcrIIIC1). In addition, if CRISPRi is employed for gene regulation, one would usually want to avoid any risk of unintended DSB induction and would therefore prefer to use well-established dCas9 systems.

The active Cas9:AcrIIIC1 chimera would be more desirable than the traditional dCas9 system in gene circuits that require easy alternation between cleavage and binding activity. In the case of the Cas9:AcrIIIC1 chimera, this would be achieved by simply inducing or not inducing the expression of AcrIIIC1, while the conventional CRISPRi system would necessitate an additional active *cas9* gene as well as strictly control regulation of both dCas9 and Cas9. In our study, we provide examples of such gene circuits by developing Acr base-editors that couple base-editing applications (by allowing AcrIIIC1 expression) to subsequent counter-selection (by interrupting AcrIIIC1 expression). The applicability of these circuits could be further expanded for the study of microbial communities, which are typically composed of non-model organisms, providing an easy and flexible way to control population dynamics. Transient expression of AcrIIIC1 could be used to control the growth rate and/or productivity of certain community members, without the requirement for use of a tight expression system. Controlled reduction in the AcrIIIC1 expression would eliminate these members from the community, allowing to study the effect of the population changes to the community. For clarity, we added this example in the Discussion section ('In addition, the applicability of our active Cas9:Acr systems for gene targeting inhibition, gene silencing and base-editing could be further expanded for the study of microbial communities, which are typically composed of non-model organisms, providing an easy and flexible way to control population dynamics. Transient expression of AcrIIIC1 could be used to control the growth rate and/or productivity of certain community members, without the requirement for use of a tight expression system. Controlled reduction in the AcrIIIC1 expression would eliminate these members from the community, allowing to study the effect of the population changes to the community.').

Beyond these critics regarding novelty and applicability of the presented CRISPRi approach, I do acknowledge the utility AcrIIIC1 in context of base editing showcased by the authors. My understanding of this system is that AcrIIIC1 here serves as a somewhat leaky inhibitor of Cas9 DNA cleavage. Hence, when base editing occurs sufficiently quick and close to the PAM, the resulting cells are protected from cell killing via Cas9-induced DSBs. This is, indeed, a nice strategy taking advantage of the AcrIIIC1 inhibitory mechanism and the Acr's "leakiness" regarding Cas9 inhibition.

On the experimental side, the study is very much focused on genomically-integrated GFP as a target. The study would certainly benefit from characterization of their system at endogenous sites, e.g. in the context of the interesting base-editing work the authors have done. I also invite the authors to think about applications that are only possible with their AcrIIC1-based approaches, e.g. in the context of base editing, which have not been possible before. An experimental demonstration of such a biological application would greatly strengthen this paper. Also, the paper would be considerably strengthened, if the authors could showcase their AcrIIC1 base editing strategy in another organism, in particular in mammalian cells.

A major strength of our anti-CRISPR/Cas base-editing tool is the use of the 'leaky' counter-selection nuclease (i.e. Cas9 inhibited by the anti-CRISPR) that kills non-edited cells, as acknowledged by the reviewer. We feel we indeed haven't emphasized this aspect enough in our study. As such, we pointed this out more clearly in the Results section 'AcrIIC1:Cas9 complexes as alternatives of dThermocas9 and dGeocas9 for base-editing in *E. coli*' by adding the sentence 'Hence, these Acr base-editors would first induce base-editing of the target site by expressing AcrIIC1 that blocks the Cas9 nuclease activity of the Cas9-PmCDA1 fusion protein and allows for PmCDA1-mediated deamination of the target base(s). Subsequent interruption of the AcrIIC1 expression would result in Cas9-mediated counter-selection of the wild-type cells.'. Furthermore, the added value of our Acr base editors is that they offer higher editing purity (higher number of clean mutants) and a higher frequency of mutagenesis (number of nucleotides that are simultaneously mutated) compared to the 'canonical' dCas9 base editors. Taken together, we hope to have shown that our tool offers substantial benefits over current technologies.

Furthermore, in the revised version of our manuscript, we have applied both the dCas9 and the Acr base-editors not only for the genomically-integrated GFP but also for three different endogenous sites (*pyrE*, *xytB* and *adhE* genes). These additional results have been integrated into the manuscript as Supplementary Figures 11 and 12. Moreover, the Results section 'AcrIIC1:Cas9 complexes as alternatives of dThermocas9 and dGeocas9 for base-editing in *E. coli*', the Methods section 'Base-editing assays' and the Supplementary Tables 2, 3 and 4 have been adjusted accordingly.

Finally, our base-editors are not directly translatable to mammalian cells, due to the difference in repair pathways preferences between prokaryotes and human cells. Especially the prevalence of NHEJ (non-homologous end-joining) in human cells will diminish the benefits that our 'leaky' counter-selection approach will have over more conventional approaches. Contrary to base-editing applications in eukaryotes that predominantly use nickase Cas9 variants, our base-editors are dedicated to applications in prokaryotes.

Finally, the word "in vivo" in the title of the paper is somewhat imprecise, since in the context of CRISPR it often refers to use in model organisms such as mice. I am aware that different communities use the term "in vivo" differently, but I would still suggest specifying in the title that the characterization was done in *E. coli*.

We agree with the reviewer that this might be confusing to some readers. Hence, the title '*In vivo* characterization of the AcrIIC1 anti-CRISPR protein for Cas9-based genome engineering' was replaced by 'Characterization of the AcrIIC1 anti-CRISPR protein for Cas9-based genome engineering in *E. coli*'.

I hope the authors find my comments useful and can understand my reasoning for not supporting publication in Nature Communications. That being said, I think this is an interesting study overall and certainly a good candidate for publication in a more specialized journal.

We truly appreciate the constructive criticism on our manuscript. We believe it has only strengthened our paper even further.

Reviewer #2 (Remarks to the Author):

The manuscript by Mougiakos et al., describes the genome editing and gene silencing technologies using thermostable CRISPR/Cas9 systems and its anti-CRISPR systems. The authors investigated the *in vivo* activity of previously identified thermostable Cas9s from *Geobacillus thermodenitrificans* T12 and *Geobacillus stearothermophilus* in *E. coli*. They also demonstrate the inducible blocking of these nuclease by expressing anti-CRISPR protein from *Neisseria meningitidis*. Although this is the first report on the simultaneous use of the type II-C Cas9 proteins and their Anti-CRISPR proteins *in vivo*, these proteins have been already investigated in detail previously (Ref. 34-37). Thus, this reviewer strongly recommend submission of this manuscript to a more specialized journal.

We would like to thank Reviewer #2 for the feedback. The references 34-37 describe the *in vitro* characterization of ThermoCas9, GeoCas9 and AcrIIC1, but we indeed have now characterized these proteins in *E. coli*. We would also like to point the reviewer to our replies to reviewer #1, where we clarify the novelty and benefits of our tool over existing technologies.

Minor points;

- Line 76, Please spell out 'REC'.

The phrase 'The more compact REC lobe of II-C Cas9 proteins...' was replaced by 'The more compact recognition (REC) lobe of II-C Cas9 proteins...'.

- In Fig. 2a, gfp7, the bar color is confusing.

The indicated bar color in Fig. 2a was corrected.

Reviewer #3 (Remarks to the Author):

In this manuscript, the authors used "In vivo characterization of the AcrIIC1 anti-CRISPR protein for Cas9-based genome engineering", which imply to audience that the content of the study should be focused on the AcrIIC1. However, the authors made great efforts to characterize thermostable type II-C ThermoCas9 and GeoCas9 endonucleases in genome editing for their activity of introducing lethal DSDBs and PAM recognition sequences *in vivo* at 37°C in the *E. coli* genome, and achieved almost the same conclusion as the authors reported previously *in vitro* 36. Therefore, even for characterization of ThermoCas9 and GeoCas9, no new finding could be claimed.

The authors prove that the previously *in vitro* characterized anti-CRISPR protein (AcrIIC1) robustly inhibits the DNA cleavage activity of both ThermoCas9 and GeoCas9 *in vivo*, thus claiming that "this is the first time that an Acr protein is shown to interact with ThermoCas9, enabling applications that require 'off-switch' control of its cleavage activity". Given that AcrIIC1 is able to inhibit the DNA cleavage activity of Cas9-mediated editing activity has been well known in gene editing community 34,35, this finding is not surprised. Therefore, the two claims, in which ThermoCas9 and GeoCas9 can

be used as efficient genome targeting or editing tools for in vivo applications at 37°C, as well as have the ability of 'off-switch' control by AcrIIC1, are lack of important novelty.

We would like to thank Reviewer #3 for the feedback. Reviewer #1 raised the same issue. Please see our second response above on this matter.

The authors use AcrIIC1:Cas9 complexes to uncover the ability of AcrIIC1 to trap the mutant Cas9 nucleases in a DNA-bound and cleavage-inactive state in vivo, truly resulting in silencing of the targeted gfp gene in the genome of E. coli, but the silencing efficiency was lower than that of their dCas9 variants, this means that the strategy using AcrIIC1:Cas9 complexes is not a good alternative for replacement of dCas9 variants, which have already been extensively used in gene editing community.

We address this issue in our 4th response to Reviewer #1.

The authors also develop AcrIIC1:Cas9 and dCas9 base-editors of the two II-C variants and observe up to six times larger editing windows than the previously reported dSpyCas9 base-editor. The dThermoCas9 and dGeoCas9 editors mainly edit the PAM-distal end of the protospacer, while the Acr-editors associated to the active Cas9 variants show preference for editing at the PAM-proximal end of the protospacer. The extended base-editing windows of all II-C editor designs described in this study may increase the freedom for gene inactivation, for abrogation of regulatory sequences and splice sites. These are main findings with novelty, which would providing some useful information to gene editing community.

We appreciate the positive remarks on our study.